# Impact of Pneumococcal Urinary Antigen Testing in COVID-19 Patients: Outcomes from the San Matteo COVID-19 Registry (SMACORE)

**DOI:** 10.3390/jpm11080762

**Published:** 2021-07-31

**Authors:** Pietro Valsecchi, Marta Colaneri, Valentina Zuccaro, Erika Asperges, Filippo Costanzo, Bianca Mariani, Silvia Roda, Rita Minucci, Francesco Bertuccio, Elia Fraolini, Matteo Bosio, Claudio Tirelli, Tiberio Oggionni, Angelo Corsico, Raffaele Bruno

**Affiliations:** 1Division of Infectious Diseases I, Fondazione IRCCS Policlinico San Matteo, 27100 Pavia, Italy; pietro.valsecchi01@universitadipavia.it (P.V.); marta.colaneri01@universitadipavia.it (M.C.); v.zuccaro@smatteo.pv.it (V.Z.); erika.asperges01@universitadipavia.it (E.A.); silvia.roda01@universitadipavia.it (S.R.); 2Division of Internal Medicine, Fondazione IRCCS Policlinico San Matteo, 27100 Pavia, Italy; f.costanzo@smatteo.pv.it; 3Unit of Microbiology and Virology, Fondazione IRCCS Policlinico San Matteo, 27100 Pavia, Italy; b.mariani@smatteo.pv.it; 4Faculty of Medicine, University of Pavia, 27100 Pavia, Italy; rita.minucci01@universitadipavia.it; 5Division of Chest Medicine, IRCCS Policlinico San Matteo Foundation, 27100 Pavia, Italy; francesco.bertuccio01@gmail.com (F.B.); e.fraolini@smatteo.pv.it (E.F.); ma.bosio@smatteo.pv.it (M.B.); c.tirelli@smatteo.pv.it (C.T.); t.oggionni@smatteo.pv.it (T.O.); angelo.corsico@unipv.it (A.C.); 6Department of Clinical, Surgical, Diagnostic, and Paediatric Sciences, University of Pavia, 27100 Pavia, Italy

**Keywords:** COVID-19, SARS-CoV-2, *Streptococcus pneumoniae*, pneumococcal urinary antigen, antibiotic therapy

## Abstract

Despite low rates of bacterial co-infections, most COVID-19 patients receive antibiotic therapy. We hypothesized that patients with positive pneumococcal urinary antigens (PUAs) would benefit from antibiotic therapy in terms of clinical outcomes (death, ICU admission, and length of stay). The San Matteo COVID-19 Registry (SMACORE) prospectively enrolls patients admitted for COVID-19 pneumonia at IRCCS Policlinico San Matteo, Pavia. We retrospectively extracted the data of patients tested for PUA from October to December 2020. Demographic, clinical, and laboratory data were recorded. Of 469 patients, 42 tested positive for PUA (8.95%), while 427 (91.05%) tested negative. A positive PUA result had no significant impact on death (HR 0.53 CI [0.22–1.28] *p*-value 0.16) or ICU admission (HR 0.8; CI [0.25–2.54] *p*-value 0.70) in the Cox regression model, nor on length of stay in linear regression (estimate 1.71; SE 2.37; *p*-value 0.47). After adjusting for age, we found no significant correlation between urinary antigen positivity and variations in the WHO ordinal scale and laboratory markers at admission and after 14 days. We found that a positive PUA result was not frequent and had no impact on clinical outcomes or clinical improvement. Our results did not support the routine use of PUA tests to select COVID-19 patients who will benefit from antibiotic therapy.

## 1. Introduction

Bacterial co-infection is frequently described in patients affected by viral respiratory infections and is characterized by increased morbidity and mortality. During the 2009 H1A1 influenza pandemic, bacterial co-infection was reported in 12% of hospitalized patients, with the proportion growing to 30% in those admitted to intensive care units (ICU). *Staphylococcus aureus* and *Streptococcus pneumoniae* were the most frequent co-pathogens [1,2]. The frequency of bacterial co-infections in COVID-19 patients varies widely in different studies but seems to be low globally. A systematic review and meta-analysis reported an overall prevalence of bacterial infections of 7.1% [3,4,5,6,7]. Interestingly, the prevalence of bacterial co-infections upon presentation was lower than the rate of secondary infections occurring more than 48 h after hospital admission (3.5% versus 15.5%).

Despite these low rates of co-infections, the majority of COVID-19 patients (more than 70%) receive empirical antibiotic therapy for community-acquired pneumonia [8]. In order to optimize and limit antibiotic use according to antibiotic stewardship principles, some authors have suggested the use of the pneumococcal urinary antigen (PUA) testing [9,10]. In fact, although the rates of pneumococcal and SARS-CoV-2 co-infection seems to be low, *S. pneumoniae* is one of the most frequent co-pathogens in respiratory viral infections, and a PUA test is a non-invasive and cost-effective tool, with proven efficacy in the diagnosis of community-acquired pneumonia (CAP) [10,11].

We hypothesized that COVID-19 patients with a positive PUA result may benefit from appropriate antibiotic treatment in terms of mortality, ICU admission, and length of stay in comparison to those who tested negative. Furthermore, we analyzed clinical improvement between patients with positive PUA results and those with negative PUA results.

## 2. Methods

### 2.1. Patients and Study Design

The San Matteo COVID-19 Registry (SMACORE) cohort has collected records on COVID-19 patients hospitalized in our hospital (IRCCS Policlinico San Matteo, Pavia, Italy) since the beginning of the pandemic (February 2020). Confirmed COVID-19 pneumonia was defined as radiological findings of pneumonia plus SARS-CoV-2 positivity on a nasopharyngeal swab analyzed with real-time polymerase chain reaction (RT-PCR) by our Molecular Virology Unit. Data on demographics, clinic, biochemical and microbiologic analyses, treatment, and outcome (admission to ICU, death, and discharge) are included. Data collection is still ongoing.

Ethical approval for observational research using SMACORE data was obtained from the local ethics committee.

In this study, we retrospectively extracted medical records from the SMACORE cohort for patients admitted between October 2020 and December 2020 and abstracted anonymized data from standardized data collection forms. This period of recruitment was chosen because a new test to detect urinary antigens was implemented in our laboratory at the beginning of October 2020. Demographic data included age and sex; clinical data included medical history, the Charlson comorbidity index (CCI), the onset of symptoms, steroid and antibiotic therapy prior to hospitalization, antibiotic therapy during hospitalization, and clinical status expressed with the WHO ordinal scale; laboratory findings included C-reactive protein levels, procalcitonin levels, and the neutrophil and lymphocyte count at admission and day 7 [12,13]. Radiological findings were divided into four main groups according to radiological reports: ground-glass opacities (GGO), focal opacities, both focal and ground-glass opacities, and other radiological findings [13].

We divided our cohort of COVID-19 patients into two groups according to the PUA test result, and then compared them to assess the differences in primary and secondary outcomes.

The primary outcome of the study was the 30-day in-hospital mortality rate. Secondary outcomes were ICU admission and clinical improvement according to the WHO ordinal scale at the 7th and 14th day after admission.

### 2.2. Microbiological Methods

The urine samples were collected and processed in our microbiology laboratory.

PUAs were detected with STANDARD TM F *S. pneumoniae* Ag FIA (SD Biosensor Inc., Republic of Korea), a fluorescence immunoassay. The test line of the assay is coated with rabbit anti *S. pneumoniae* capsular polysaccharides (CPS). The patient’s sample is applied to the test device, where it migrates through a membrane. If PUA is present, it will react with europium conjugated monoclonal anti-*S. pneumoniae* CPS in the conjugation pad and form antigen–antibody fluorescence particle complexes. The complexes move along the membrane to be captured by the rabbit anti *S. pneumoniae* CPS on the test line and form a fluorescence signal. The analyzer using pre-programmed algorithms scans the intensity of the fluorescence light. The analyzer indicates a Cutoff Index value (COI). A COI ≥ 1 is considered as a positive result. The test has shown a sensitivity of 80.8% and a specificity of 84.8% in patients aged >17 with cultural evidence of *S. pneumoniae* infection.

### 2.3. Statistical Analysis

Data for continuous variables are presented as means and standard deviations. Data for categorical variables are presented as frequencies and percentages. Comparisons between positive PUA results and negative PUA results were performed using chi-square tests for categorical variables and *t*-tests for continuous variables.

The incidence, mechanisms, and patterns of missing data were then explored. Since the amount of missing data was non-negligible (higher than 10% in multiple variables) and followed a Missing at Random Mechanism, multiple imputations with predictive mean matching (20 datasets, 50 iterations) were employed to reduce their impact. The quality of imputed data was ascertained by comparing the distributions of imputed datasets with the ones with complete data. Multiple imputed datasets were then employed to perform the subsequent analyses.

First, two multivariable Cox proportional-hazard regression models were used to assess the predictive effect of a positive PUA result on 30-day mortality and ICU admission. Each multivariable model was developed including variables that were associated with the dependent variable in bivariate Cox regressions.

Second, the predictive effect of a positive PUA result on length of stay (LOS) in patients who survived after the 30th day was investigated. To achieve this, a multivariable linear regression was used. Similarly, the multivariable model included the demographic, clinical, radiological, and laboratory variables which were significantly associated with LOS in bivariate analyses.

Finally, linear mixed models were performed to evaluate possible differences between positive and negative PUA patients in the variance of biochemical values (C reactive protein, neutrophils, and lymphocytes) and in the WHO ordinal scale (the first expressed as the difference between day 0 and day 7; the second as the difference between day 0 and day 7 and between day 0 and day 14). Time, PUA, and their interaction were included as fixed effects. A significant time–PUA interaction indicates that there was a difference between the two groups in the variance of biochemical values and the WHO ordinal scale. The fixed effect of age was also included in the models to account for confounding.

Results of the Cox proportional-hazard regressions are reported as Hazard Ratios (HR) with 95% Confidence Intervals (CI); results of the linear regression and the linear mixed models are reported as unstandardized betas. The significance threshold was set at α = 0.05.

## 3. Results

### 3.1. Study Population

The main characteristics of the 469 patients included in our cohort were as follows: 320 were male (68.2%) and 149 were female (31.8%), with a median age of 67.6 years (SD +/−14.7). Forty-two (8.95%) patients tested positive for PUA, while 427 (91.05%) tested negative. The characteristics of the study population are shown in Table 1. Patients with positive PUA results showed significantly higher levels on the WHO ordinal scale at admission and after 7 days, but not after 14 days. Patients with positive PUA results more frequently showed focal opacities on the chest X-ray, and less frequently showed ground-glass opacities alone, in comparison to patients with negative PUA results. The multiple imputation procedure converged, and imputed data followed plausible distributions. The imputed datasets were therefore employed in the following analyses.

### 3.2. Effects of Positive Urinary Antigen on Mortality and ICU Admission

Age, CCI, severe immunosuppression, days since the onset of symptoms, the WHO ordinal scale at admission, and CRP levels were statistically significant in predicting 30-day mortality in bivariate Cox regression models and were included in the multivariable model. A positive PUA result was not significantly associated with 30-day mortality in either the bivariate or the multivariable Cox regression models (Table 2). Regarding ICU admission, procalcitonin levels, previous antibiotic treatment, and CCI were significant in the bivariate analyses, whereas a positive PUA result was not significantly associated with ICU admission in the bivariate and the multivariable models (Table 2; Figure 1).

### 3.3. Effects of Positive Urinary Antigen on Length of Stay

The procalcitonin levels at admission, the presence of both GGOs and FOs in CT scans, the WHO ordinal scale at admission, and antibiotic treatment with three different antibiotics during hospitalization were significant in predicting length of stay (LOS) in the bivariate analysis and were therefore included in the multivariate linear regression model. A positive PUA result did not show any predictive value for LOS in the bivariate analysis and multivariate linear regression (Table 3).

### 3.4. Effects of Positive Urinary Antigen on Variation of WHO Ordinal Scale and Laboratory Measures

Patients with positive PUA results did not show a different variation on the WHO ordinal scale after 7 and 14 days compared with patients with negative PUA results in linear mixed models (Table 4; Figure 2). After being corrected for age, no significant variance was found in CRP levels, or in the neutrophil and lymphocyte count between admission and after 7 days in the two PUA groups (Table 5; Figure 2).

## 4. Discussion

The appropriateness of antibiotic use is an important topic in the context of COVID-19 management in the era of antibiotic stewardship programs. In this study, we retrospectively analyzed the impact of PUA positivity on clinical outcomes in a cohort of COVID-19 patients. We did not observe any impact of a positive PUA result on clinical improvement or clinical outcomes in terms of mortality, ICU admission, and length of stay. Our results were similar to Desai et al.’s work, which showed no impact on mortality and length of stay in COVID-19 patients with a positive PUA result in the emergency department setting [14].

Compared to influenza pandemics, our findings show a significantly lower rate of *Streptococcus pneumoniae* co-infection in COVID-19 patients [2]. This result is consistent with those of other studies [6,7,14]. Potential reasons for such a low occurrence of co-infection with other respiratory pathogens might reside in the implementation of measures such as social distancing, universal masking, and the extended use of personal protection equipment [15,16,17,18]. According to the 2019 Infectious Disease Society of America (IDSA) guidelines for CAP, urinary antigen testing for *Streptococcus pneumoniae* and *Legionella pneumophila* is recommended only in patients who meet the criteria for severe CAP [19]. Severe COVID-19 patients often meet these criteria and are treated with empirical broad-spectrum antibiotics [6,20]. PUA has been proposed as a tool to optimize antibiotic prescription and select patients who may benefit from antibiotic therapy [9]. Our results did not show any benefit in terms of clinical outcomes, but we think that the PUA test could be a useful tool in COVID-19 patients when used not as a routine test but rather when a rational approach is applied according to the principles of antimicrobial and diagnostic stewardship. In fact, routine testing may result in increased rates of false-positive results and consequently lead to inappropriate treatment, since urinary antigen excretion may persist after previous pneumococcal pneumonia and pneumococcal vaccination [21,22]. Moreover, false positives can occur in the presence of proteinuria, which is frequent in COVID-19 [23].

Our work has several limitations. First, due to the retrospective nature of this study, missing data were frequent, but this was accounted for in the analyses. Second, since we only enrolled patients in our center over a small interval of time, we had a limited sample. Furthermore, we did not investigate the presence of other bacterial co-infections through cultures from blood and respiratory samples, which may have helped us exclude cases of false positives due to test hypersensitivity. Regarding blood cultures, the sample collection was heterogenous from unit to unit, therefore the possible results were difficult to interpret. A sputum culture was not routinely performed, since COVID-19 patients usually present with a non-productive cough. Furthermore, aerosol-generating procedures such as sputum induction increase the risk of SARS-CoV-2 transmission and were therefore limited to selected cases. Further studies can provide evidence concerning the impact that this approach could have on clinical outcomes and antimicrobial consumption in COVID-19.

## Figures and Tables

**Figure 1 jpm-11-00762-f001:**
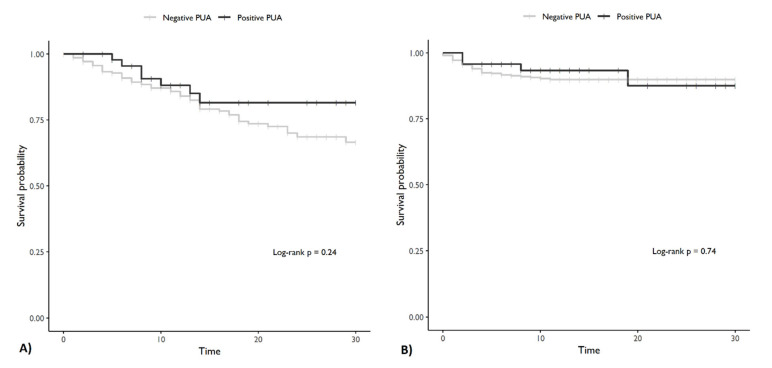
Survival curves. (**A**) Survival curve for death. (**B**) Survival curve for ICU admission. Note: PUA = pneumococcal urinary antigen.

**Figure 2 jpm-11-00762-f002:**
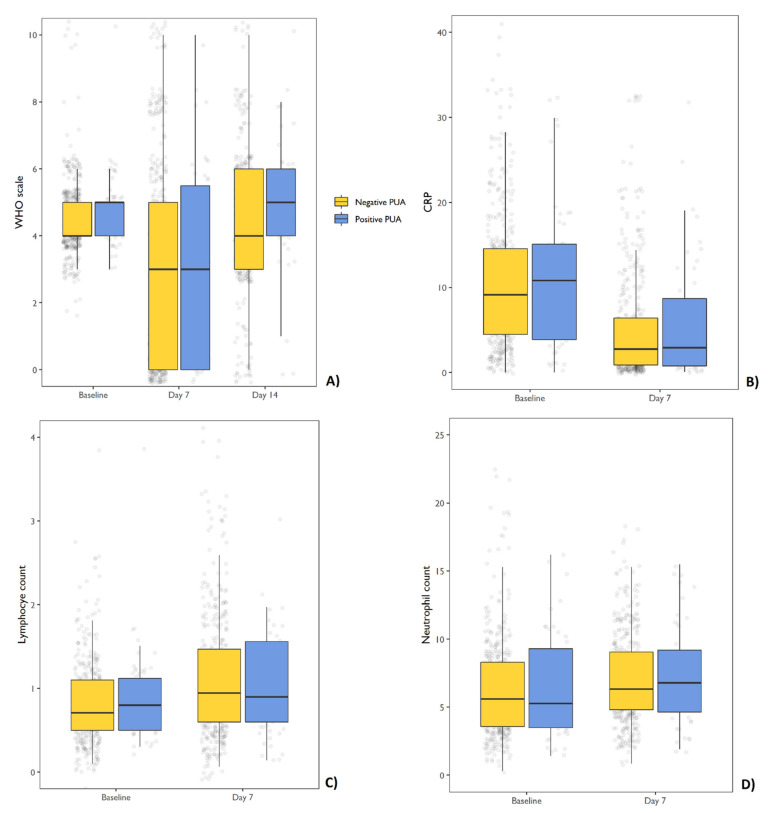
Box and whiskers charts. (**A**) Variation in WHO ordinal scale at 7 and 14 days from baseline. (**B**) Variations in CRP levels at 7 days from baseline; (**C**) variations in the neutrophil count at 7 days from baseline; (**D**) variation of the lymphocyte count at 7 days from baseline. Abbreviations CRP C-reactive protein, PUA pneumococcal urinary antigen.

**Table 1 jpm-11-00762-t001:** Demographic and clinical characteristics.

Variable		Whole Sample (n = 469)	Negative PUA (n = 427)	Positive PUA (n = 42)	*p*-Value
Sex, n (%)	Male	320 (68.2)	295 (69.1)	25 (59.5)	
	Female	149 (31.8)	132 (30.9)	17 (40.5)	0.27
Age, n (%)	mean (sd)	67.6 (14.7)	67.6 (14.7)	67.6 (14.5)	0.98
Death, n (%)	yes	84 (17.9)	79 (18.5)	5 (11.9)	
	no	385 (82.1)	348 (81.5)	37 (88.1)	0.39
ICU admission, n (%)	yes	43 (9.2)	40 (9.4)	3 (7.1)	
	no	426 (90.8)	387 (90.6)	39 (92.9)	0.84
CCI, M (sd)		3.5 (2.3)	3.4 (2.3)	3.8 (2.3)	0.27
	missing	43	43	0	
Severe immunodepression, n (%)		13 (3.0)	11 (2.8)	2 (4.9)	0.8
	missing	39	38	1	
Previous lung disease, n (%)	No	228 (64.0)	205 (64.3)	23 (62.2)	
	Yes	128 (36.0)	114 (35.7)	14 (37.8)	0.94
	missing	113	108	5	
Days since onset of symptoms, M (sd)		6.3 (3.8)	6.2 (3.8)	6.6 (3.9)	0.53
	missing	65	63	2	
Steroid treatment before hospitalization, n (%)	No	309 (74.5)	281 (75.1)	28 (68.3)	
	Yes	106 (25.5)	93 (24.9)	13 (31.7)	0.44
	missing	54	53	1	
Antibiotic treatment before hospitalization (n of atb), n (%)	0	281 (67.2)	250 (66.5)	31 (73.8)	
	1	125 (29.9)	117 (31.1)	8 (19.0)	
	2	12 (2.9)	9 (2.4)	3 (7.1)	0.07
	missing	51	51	0	
Antibiotic treatment during hospitalization (n of atb), n (%)	0	18 (4.7)	16 (4.7)	2 (5.1)	
	1	254 (66.7)	227 (66.4)	27 (69.2)	
	2	103 (27.0)	94 (27.5)	9 (23.1)	
	3	6 (1.6)	5 (1.5)	1 (2.6)	0.9
Radiological findings at CT scan	No CT scan	256 (71.9)	228 (71.7)	28 (73.7)	
	GGO	21 (5.9)	20 (6.3)	1 (2.6)	
	FO	8 (2.2)	7 (2.2)	1 (2.6)	
	GGO+FO	69 (19.4)	61 (19.2)	8 (21.1)	
	others	2 (0.6)	2 (0.6)	0 (0.0)	0.89
	missing	113	109	4	
Radiological findings at chest X-ray	0	17 (4.8)	17 (5.3)	0 (0.0)	
	GGO	126 (35.3)	120 (37.6)	6 (15.8)	
	FO	55 (15.4)	45 (14.1)	10 (26.3)	
	GGO+FO	151 (42.3)	130 (40.8)	21 (55.3)	
	others	8 (2.2)	7 (2.2)	1 (2.6)	0.02
	missing	112	108	4	
WHO ordinal scale at admission	mean (sd)	4.5 (1.1)	4.5 (1)	4.9 (1.2)	0.01
	missing	46	46	0	
WHO ordinal scale at day 7	mean (sd)	4.4 (1.9)	4.3 (1.9)	5 (1.9)	0.02
	missing	70	69	1	
WHO ordinal scale at day 14	mean (sd)	3.1 (2.7)	3.1 (2.7)	3.2 (2.7)	0.91
	missing	144	140	4	
C-reactive protein at admission (mg/mL), M (sd)		11 (8.6)	10.9 (8.6)	12.5 (9)	0.28
	missing	42	38	4	
Procalcitonin at admission (ng/mL), M (sd)		1.1 (5.8)	1.1 (6)	1.4 (3.7)	0.76
	missing	169	160	9	
Neutrophil count at admission (10^3^/mm^3^), M (sd)		36.9 (440.8)	39.8 (461.6)	6.7 (3.6)	0.66
	missing	49	44	5	
Lymphocyte count at admission (10^3^/mm^3^), M (sd)		9.9 (101.7)	10.6 (106.5)	1.7 (5.2)	0.6
	missing	47	42	5	
C-reactive protein at day 7 (mg/mL), M (sd)		5 (6.3)	5 (6.2)	5.5 (7.6)	0.69
	missing	211	196	15	
Neutrophil count at day 7 (10^3^/mm^3^)		10.4 (52)	10.6 (54.8)	7.8 (3.8)	0.79
	missing	208	192	16	
Lymphocyte count at day 7 (10^3^/mm^3^), M (sd)		1.2 (1.6)	1.3 (1.7)	1.1 (0.5)	0.52
	missing	208	192	16	

Note. Abbreviations: PUA, pneumococcal urinary antigen; M, mean; SD, standard deviation; ICU, intensive care unit; CCI, Charlson comorbidity index; GGO, ground-glass opacity; FO, focal opacity.

**Table 2 jpm-11-00762-t002:** Multivariate Cox regression for 30-day mortality and ICU admission.

Variable	HR	95% CI	*p*-Value
30 days mortality			
Age	1.09	[1.06–1.12]	<0.001
CCI	0.98	[0.84–1.14]	0.78
Severe immunosuppression	5.87	[0.87–1.03]	0.01
Days since onset of symptoms	0.95	[0.87–1.03]	0.20
WHO ordinal scale at admission	1.21	[1.02–1.43]	0.04
CRP levels (mg/mL)	1.02	[1–1.04]	0.03
Positive PUA	0.53	[0.22–1.28]	0.16
ICU admission			
CCI	0.85	[0.73–0.99]	0.05
Antibiotic treatment before hospitalization (atb = 1)	2.2	[1.12–4.35]	0.03
Antibiotic treatment before hospitalization (atb = 2)	3.67	[1.06–12.75]	0.05
PCTI levels (ng/mL)	1.04	[1.01–1.07]	0.01
Positive PUA	0.8	[0.25–2.54]	0.70

Note. Abbreviations: HR, Hazard Ratio; CI, Confidence Interval; CCI, Charlson comorbidity index; CRP, C-reactive protein; atb, antibiotic; PCTI, procalcitonin; PUA, pneumococcal urinary antigen.

**Table 3 jpm-11-00762-t003:** Multivariate linear regression for length of stay.

Variable	Estimate	SE	*p*-Value
Intercept	−3.64	5.26	0.49
Antibiotic treatment before hospitalization (atb = 1)	0.29	1.56	0.85
Antibiotic treatment before hospitalization (atb = 2)	7.24	4.07	0.08
Antibiotic treatment during hospitalization (atb = 1)	−2.13	3.57	0.55
Antibiotic treatment during hospitalization (atb = 2)	−2.31	3.67	0.53
Antibiotic treatment during hospitalization (atb = 3)	19.02	6.62	<0.001
WHO ordinal scale at admission	3.51	0.81	<0.001
CT scan GGO	5.64	3.23	0.08
CT scan FO	0.45	5.48	0.93
CT scan GGO and FO	10.17	1.93	<0.001
CT scan others	4.11	8.49	0.63
Chest X-ray GGO	1.83	3.52	0.60
Chest X-ray FO	4.36	3.74	0.25
Chest X-ray GGO and FO	2.93	3.63	0.42
Chest X-ray others	4.29	6.53	0.51
CRP at admission (mg/mL)	0.14	0.09	0.13
PCTI at admission (ng/mL)	0.49	0.19	0.01
Positive PUA	1.71	2.37	0.47

Note. Abbreviations: SE, standard error; GGO, ground-glass opacity; FO, focal opacity; CRP, C-reactive protein; PCTI, Procalcitonin, PUA pneumococcal urinary antigen.

**Table 4 jpm-11-00762-t004:** Linear mixed models for WHO ordinal scale variation at day 7 and 14.

Variable	Estimate	SE	*p*-Value
Intercept	2.30	0.38	<0.001
Time (day 7 vs. day 0)	−0.10	0.12	0.41
Time (day 14 vs. day 0)	−1.33	0.14	<0.001
PUA	0.37	0.32	0.26
Age	0.03	0.01	<0.001
Time (day 7 vs. day 0) × PUA	0.19	0.36	0.60
Time (day 14 vs. day 0) × PUA	−0.38	0.39	0.34

Note. Abbreviations: SE, standard error; PUA, pneumococcal urinary antigen.

**Table 5 jpm-11-00762-t005:** Variation in laboratory findings at 7 days from baseline in linear mixed models.

Variable	C-Reactive Protein (mg/mL)			Lymphocyte Count (10^3^/mm^3^)			Neutrophil Count (10^3^/mm^3^)		
	est	SE	*p*-Value	est	SE	*p*-Value	est	SE	*p*-Value
Intercept	6.33	1.49	<0.001	148.12	48.39	<0.001	3652	10.80	<0.001
Time (day 7 vs. day 0)	−5.87	0.54	<0.001	−19.89	21.74	0.36	−8.55	4.70	0.07
PUA	1.01	1.27	0.43	−31.22	45.30	0.49	−8.61	10.25	0.40
Age	0.07	0.02	<0.001	−1.64	0.68	0.02	−0.39	0.15	0.01
Time (day 7 vs. day 0) × PUA	−0.59	1.52	0.70	26.36	65.15	0.69	8.16	14.46	0.57

Note. Abbreviations: PUA, pneumococcal urinary antigen.

## Data Availability

The data presented in this study are available on request from the corresponding author. The data are not publicly available due to privacy restrictions.

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
