# Peer review of "Impact of Pneumococcal Urinary Antigen Testing in COVID-19 Patients: Outcomes from the San Matteo COVID-19 Registry (SMACORE)"

_jpm, 2021, doi:10.3390/jpm11080762_

Round 1
Reviewer 1 Report
The gathering the latest international scientific findings and knowledge on COVID-19 is the issue of interest during current times. All observations made during course of disease enrich the awareness of different aspects of the pandemic and help us to better prepare for the next unforeseen epidemic. The manuscript puts some data on that list.
I have no objection to the quality of the manuscript. There are only a few points of attention for minor corrections:
1. Technically, p valuescannot equal 0. Some statistical programs do give you p values of 0.000 in their output, but this is likely due to automatic approximation or truncation to a preset number of digits after the decimal point. So, consider replacing "p =0 .000" with "p <0 .0001," since the latter is considered more acceptable and does not substantially alter the importance of the p value reported.
2. The second issue is to change coma to decimal points in Tables.
3. In Table 5, I suppose that by mistake "lymphocyte count" is written twice in columns.
Author Response
Thank you for your kind revision, I'll apport all the corrections you have suggested to the manuscript
Reviewer 2 Report
As the authors mentioned, the frequency of bacterial co-infections in COVID-19 patients seems to be generally low. The use of antibiotics in patients with COVID-19 pneumonia did not respond to the evidence of their bacterial infection. Optimizing antibiotics usage is essential. Here are my suggestions for the manuscript:
- The aim of the manuscript is to explore the association between the pneumococcal urinary antigen (PUA) and the clinical outcomes of the patients. What does PUA stand for in this study? It represents neither the status of antibiotics usage (in Table1, the groups of positive and negative PUA had similar antibiotics usage) nor evidence of bacterial infection (no blood or sputum culture data in the study).
- According to the IDSA guidelines and the references in the manuscript, PUA should be obtained with the blood and sputum cultures of the patients simultaneously. Please supplement available cultural evidence and make a comparison with PUA. Furthermore, PUA might be a practical tool in making the decision of antibiotics usage in culture-negative patients.
- Please describe all the significant results including the radiological findings and the WHO ordinal scales in Table 1.
- Please supplement the univariate regression models of 30-day-mortality and 30-day-ICU admission in the supplementary material.
- Please clearly describe the PUA assay utilized in the study, including its sensitivity, specificity, and reference.
Author Response
Dear reviewer, thank you for your thoughtful comments.
- Despite globally low frequency of bacterial co-infections in COVID-19, most patients receive antibiotic treatment. In our retrospective cohort we hypothesize that patients with positive PUA would benefit from antibiotic treatment more than those with negative PUA. We acknowledge that the absence of data regarding blood cultures and sputum cultures is one of the major limitations of our study.
- The IDSA guidelines for community acquired pneumonia and the recommendations from Elske Sieswerda et al. we cited in our work suggest obtaining sputum and blood sample for cultures before starting empiric antibiotic therapy alongside PUA for patients with severe CAP and for COVID-19 patients1,2 Despite being based on low quality of evidence, these suggestions seem reasonable in good clinical practice and therefore we think that your suggestion would bring added value to our work. Unfortunately, one of the biggest limitations of our study is the absence of cultural evidence. Sputum culture wasn’t routinely performed, since COVID-19 patients usually presents with non-productive cough. Furthermore, aerosol-generating procedures like sputum induction increase the risk of SARS-COV-2 transmission and where therefore limited to selected cases. We tried to retrospectively collect data regarding blood cultures at admission, but we decided not to include them in the analysis since we found that the samples collection was heterogenous from unit to unit, and therefore the possible results poorly interpretable. As we cannot provide any cultural evidence, we will underline in the discussion how this represented a major limitation of the study and the need of obtaining samples for cultural tests before starting empiric antibiotic therapy in COVID-19 patients. [1. Metlay JP, Waterer GW, Long AC, et al. Diagnosis and treatment of adults with community-acquired pneumonia. Am J Respir Crit Care Med. 2019;200(7):E45-E67. doi:10.1164/rccm.201908-1581ST. 2. Sieswerda E, de Boer MGJ, Bonten MMJ, et al. Recommendations for antibacterial therapy in adults with COVID-19 – an evidence based guideline. Clin Microbiol Infect. 2020;(xxxx):9-14. doi:10.1016/j.cmi.2020.09.041]
- We will describe the significative results from Table 1 in the text.
- We will supplement the univariate regression models of 30-days mortality and 30-days ICU admission in the supplementary material
- We will provide this information in the text.
Round 2
Reviewer 2 Report
Dear authors,
Thank you for the detailed revision. Your clarification of the study limitations and supplementation of statistical analysis improve the quality of the manuscript. It also provides us some information about PUA in diagnosing bacterial co-infection of patients with COVID-19.
This manuscript is a resubmission of an earlier submission. The following is a list of the peer review reports and author responses from that submission.